# Thrombin Induces Angiotensin II-Mediated Senescence in Atrial Endothelial Cells: Impact on Pro-Remodeling Patterns

**DOI:** 10.3390/jcm8101570

**Published:** 2019-10-01

**Authors:** Hira Hasan, Sin-Hee Park, Cyril Auger, Eugenia Belcastro, Kensuke Matsushita, Benjamin Marchandot, Hyun-Ho Lee, Abdul Wahid Qureshi, Gilles Kauffenstein, Patrick Ohlmann, Valérie B Schini-Kerth, Laurence Jesel, Olivier Morel

**Affiliations:** 1INSERM UMR1260 Regenerative NanoMedicine, Fédération de Médecine Translationnelle de Strasbourg, Université de Strasbourg, Faculté de Pharmacie, BP 60024 FR-67401 Strasbourg, France; 2Pôle d’Activité Médico-Chirurgicale Cardio-Vasculaire, Nouvel Hôpital Civil, Centre Hospitalier Universitaire, Fédération de Médecine Translationnelle de Strasbourg, Strasbourg, BP 426-67091 France

**Keywords:** senescence, thrombin, atrial fibrillation, endothelium, angiotensin II, remodeling

## Abstract

Background: Besides its well-known functions in hemostasis, thrombin plays a role in various non-hemostatic biological and pathophysiologic processes. We examined the potential of thrombin to promote premature atrial endothelial cells (ECs) senescence. Methods and Results: Primary ECs were isolated from porcine atrial tissue. Endothelial senescence was assessed by measuring beta-galactosidase (SA-β-gal) activity using flow cytometry, oxidative stress using the redox-sensitive probe dihydroethidium, protein level by Western blot, and matrix metalloproteinases (MMPs) activity using zymography. Atrial endothelial senescence was induced by thrombin at clinically relevant concentrations. Thrombin induced the up-regulation of p53, a key regulator in cellular senescence and of p21 and p16, two cyclin-dependent kinase inhibitors. Nicotinamide adenine dinucleotide phosphate NADPH oxidase, cyclooxygenases and the mitochondrial respiration complex contributed to oxidative stress and senescence. Enhanced expression levels of vascular cell adhesion molecule (VCAM)-1, tissue factor, transforming growth factor (TGF)-β and MMP-2 and 9 characterized the senescence-associated secretory phenotype of atrial ECs. In addition, the pro-senescence endothelial response to thrombin was associated with an overexpression of both angiotensin converting enzyme and AT1 receptors and was inhibited by perindoprilat and losartan. Conclusions: Thrombin promotes premature ageing and senescence of atrial ECs and may pave the way to deleterious remodeling of atrial tissue by a local up-regulation of the angiotensin system and by promoting pro-inflammatory, pro-thrombotic, pro-fibrotic and pro-remodeling responses. Hence, targeting thrombin and/or angiotensin systems may efficiently prevent atrial endothelial senescence.

## 1. Introduction

Atrial fibrillation (AF) is the most common sustained arrhythmia, especially during ageing, and portends an increased risk of thromboembolism, ischemic stroke and mortality [1]. In AF, the classic paradigm involves thrombogenesis-associated blood stasis in poorly contractile atria together with a hypercoagulable state, as witnessed by high circulating levels of fibrinolytic degradation products, plasminogen activator inhibitor, thrombin–antithrombin complex and procoagulant microparticles (MPs) levels [1,2]. Detected at high concentrations in AF [3] and in patients with ischemic stroke [4], MPs provide an additional phospholipidic surface enabling the assembly of tenase and prothrombinase complexes, ultimately leading to thrombin generation [5]. Thrombin, a serine protease central to blood coagulation, converts soluble plasma fibrinogen into insoluble clot-forming fibrin polymers. Besides its role in thrombus formation, recent findings have highlighted that thrombin conveys “non-hematological” functions through cell protease activated receptor (PAR). Activation of PAR modulates key processes involved in AF initiation and maintenance, such as inflammation and atrial remodeling, but also in altered electrical and mechanical properties of atria [6,7,8,9]. Because thrombin is short-lived in the bloodstream, most of its effects exert locally, especially on surrounding endothelial cells (ECs) [10]. At the interface between blood and atria tissue, ECs play a determinant role in the tuning of hemostatic functions even though the underlying mechanisms remain poorly investigated. Recent advances in the understanding of the purported mechanisms of thrombogenesis in cardiovascular diseases have suggested a key role for endothelial senescence. Two recent reports have highlighted the link between senescence and AF occurrence or extent of atrial fibrosis as a marker of tissue remodeling associated to AF maintenance [11,12]. Beyond fibrotic remodeling, senescence is characterized by the acquisition of a secretory phenotype. Although the major characteristic of senescence is cell-cycle arrest, recent studies revealed that cells undergoing either replicative or premature senescence acquire a senescence-associated secretory phenotype (SASP) which leads to the release of three categories of factors: inflammatory mediators, growth factors, and modifiers of the extracellular matrix (ECM) [13]. Indeed, endothelial senescent cells secrete matrix metalloproteinases MMPs, plasminogen activator inhibitor 1, growth factors, proteases, and cytokines such as IL-1, IL-6 and IL-8, all with potential strong autocrine and paracrine actions [14].

We have recently demonstrated that endothelial senescence favors thrombogenicity through several mechanisms including tissue factor (TF) up-regulation, pro-coagulant MPs release, endothelial nitric oxide synthase (eNOS) down-regulation and reduced ECs-mediated nitric oxide (NO)-dependent inhibition of platelet aggregation [15].

In the present study, taking advantage of an original model of primary atrial ECs culture, we examine whether thrombin could promote atrial endothelial senescence. In addition, the possibility that atrial endothelial senescence shifts the cell phenotype towards pro-thrombotic, pro-fibrotic, pro-inflammatory and pro-remodeling patterns was examined. Finally, since angiotensin II (AngII) via Nicotinamide adenine dinucleotide phosphate (NADPH) oxidase-derived oxidative stress is a potent inducer of premature endothelial senescence and ECs express both angiotensin-converting enzyme (ACE) and angiotensin type 1 receptors (AT1R) [15], the potential role of the local AngII/AT1R in the noxious impact of thrombin on premature endothelial senescence and dysfunction was determined.

## 2. Materials and Methods

### 2.1. Chemicals

Unless indicated, all chemicals and solvents were from Sigma-Aldrich (Sigma-Aldrich, St Quentin Fallavier, France). Human thrombin was obtained from Etablissement Français du Sang-Alsace, Strasbourg, France. Losartan was obtained from Merck Research Laboratories (NJ, Boston, MA, U.S.A) and perindoprilat from Servier (Paris, France).

### 2.2. Isolation and Culture of Atrial Endothelial Cells

Atrial ECs were isolated from left atria of porcine hearts obtained at the local slaughterhouse (SOCOPA, Holtzheim, France). Heart-lung blocks were dissected to obtain a complete left atrium. At the level of mitral valve, the left ventricle was removed, and ligation of pulmonary veins was done. The luminal surface of the left atrium was extensively flushed with phosphate-buffered saline (PBS) to discard any remaining blood. Cells were isolated by collagenase treatment (type 1, Worthington, 1 mg/mL for 40 min at 37 °C) and cultured in culture dishes containing MCDB131 medium (Invitrogen) supplemented with penicillin (100 IU/mL), streptomycin (100 IU/mL), fungizone (250 μg/mL), L-glutamine (2 mM, all from Lonza, St Quentin en Yvelines, France) and 15% fetal calf serum and grown for 3–4 days (passage 0). The medium was changed every 48 h. Endothelial phenotype was validated by von Willebrand factor and Platelet endothelial cell adhesion molecule-1 (PECAM1) staining (Additional Appendix A). Premature atrial ECs senescence was induced at passage 1 by incubating cells with either thrombin (1 or 3 U/mL) coherent with local concentration found in pro thrombotic conditions [16], or AngII (100 nM) for 24 h in serum-free medium 15 h after seeding. Pharmacological modulators were preincubated with atrial ECs, 30 min before adding thrombin with the exception of N-acetyl cysteine (NAC), which was preincubated for 3 h [17].

### 2.3. Determination of Senescence Associated-β-Galactosidase (SA-β-Gal) Activity by Flow Cytometry

SA-β-galactosidase (SA-β-gal) activity was determined by flow cytometry using the fluorogenic substrate C_12_FDG (5-dodecanoylaminofluorescein Di-β-D-galactopyranoside, Invitrogen, Life Technology, SAS) [18]. Atrial ECs were pretreated with 300 µM chloroquine for 1 h to induce lysosomal alkalinization. C_12_FDG (33 µM) was then added to the incubation medium for 1 h. At the end of the incubation period, ECs were washed with ice-cold PBS, re-suspended following trypsinization, and analyzed by flow cytometry (FACScan, BD Bioscience, San José CA, USA). Data were analyzed using Cellquest software (BD Bioscience). Light scatter parameters were used to eliminate dead cells and cellular debris. The C_12_-fluorescein signal was measured on the FL1 detector, and the proportion of ECs with SA-β-gal activity was estimated using the median fluorescence intensity of the population. Autofluorescence assessed in parallel in atrial ECs not exposed to C_12_FDG was negligible.

### 2.4. Western Blot Analysis

Atrial ECs were washed with PBS and lysed in extraction buffer of the following composition (in mM): NaCl 150, Tris/HCl 20 (pH 7.5), Na_3_VO_4_ 1, sodium pyrophosphate 10, NaF 20, okadaic acid 0.01, protease inhibitor 1X (Roche, Basel, Switzerland) and 1% Triton X-100. Proteins were separated on denaturing SDS (10%–12%) polyacrylamide gel and transferred onto nitrocellulose membranes (GE Healthcare Life Sciences, Brumath, France). Blots were blocked with 5% bovine serum albumin in PBS plus 0.1% Tween 20. Membranes were then incubated with the respective primary antibody: mouse monoclonal anti-eNOS (BD Bioscience; 610297), rabbit polyclonal anti-p53 (Santa Cruz, Heidelberg, Germany; SC-6243), mouse monoclonal anti-p21 (Santa Cruz; SC-817), rabbit polyclonal anti-p16 (Delta Biolabs; DB018), mouse monoclonal anti-tissue factor (TF, Sekisui Diagnostics, Darmstadt, Germany; 4509), rabbit polyclonal anti-VCAM-1 (Abcam, Paris, France; ab134047), mouse monoclonal anti-transforming growth factor β (TGF-β1, Abcam; ab27969), rabbit polyclonal anti-cyclooxygenase-1 (COX-1, Abcam; ab109025), rabbit polyclonal anti-COX-2 (Abcam; ab15191), rabbit polyclonal anti-angiotensin-converting enzyme (ACE, Abbiotec Nanterre, France; 250450), rabbit polyclonal anti-angiotensin type 1 receptor (AT1R, Abcam; ab124505), or mouse monoclonal anti-β-tubulin (Sigma-Aldrich; T7816) overnight at 4 °C. After washing, membranes were incubated with appropriated peroxidase-labeled secondary antibody (Cell Signaling Technology, Saint-Cyr-L’École, France; cat. nº #7074, #7076) for 60 min. Immunocomplexes were detected by chemiluminescence reaction (ECL; Amersham, Les Ulis, France) followed by densitometric analysis using the software Image J.

### 2.5. Determination of Oxidative Stress

Atrial ECs were seeded in a Millicell EZ SLIDE 8-well glass slide for 15 h, before exposure to serum-free MCDB131 (Invitrogen) for 2 h. The redox-sensitive fluorescent dye, dihydroethidium (DHE), was used to evaluate the formation of reactive oxygen species (ROS) [19]. ECs were incubated with DHE (5 μM) for 20 min at 37 °C in a light-protected manner. To determine the sources of ROS, cells were untreated or exposed to either N-acetyl cysteine (NAC, an antioxidant, 1 mM), VAS-2870 (VAS, a NADPH oxidase inhibitor, 1 μM), indomethacin (INDO, a cyclooxygenase inhibitor, 10 μM), SC-560 (a cyclooxygenase-1 inhibitor, 0.3 μM), NS-398 (a cyclooxygenase-2 inhibitor, 3 μM), mitochondrial inhibitory complex (Rotenone, 1 μM; KCN, 1 μM; Myxothiazol, 0.5 μM) before the addition of thrombin (1 U/mL) for 1 h in serum-free medium. In some experiments, ECs were incubated with either losartan (10 μM) or perindoprilat (10 μM) 30 min before the addition of thrombin (1 U/mL) for 24 h. ECs were then washed and mounted with fluorescent mounting medium (DAKO, S3023) and examined under confocal microscope (Leica SP2 UV DM Irbe). Images were analyzed using Image J software.

### 2.6. Determination of MMP Activity by Zymography

MMP-2 and MMP-9 activities in conditioned medium of cultured atrial ECs were analyzed by substrate-gel electrophoresis (zymography) with the use of SDS-PAGE (8%) containing 0.1% gelatin, as described previously [20]. Then, gels were washed in denaturing buffer for 30 min and incubated in developing buffer for 24 h at 37 °C. Thereafter, gels were stained in 30% Coomassie blue for 30 min and then de-stained with de-staining buffer. Gelatinolytic activity appeared as bands with a blue background. Images were taken and bands were analyzed using Image J software.

### 2.7. Statistical Analyses

Data are presented as Mean ± standard error of the mean (SEM) of *n* independent experiments. Mean values were compared using Student’s paired *t*-test or an analysis of variance followed by the post-hoc Bonferroni test to identify significant differences between treatments using GraphPad Prism (v5.0). The difference was considered to be significant when the *P* value was less than 0.05.

## 3. Results

### 3.1. Thrombin Induces Atrial Endothelial Cells Senescence

Thrombin, at both 1 and 3 U/mL, induced premature atrial ECs senescence, as depicted by the increased level of SA-beta-gal activity (Figure 1A). This was corroborated by the up-regulation of the key regulator in cellular senescence p53, and of p21 and p16, two cyclin-dependent kinase inhibitors (Figure 1B–D). Similar responses were observed in atrial ECs in response to AngII, a strong inducer of premature endothelial senescence (Figure 1).

### 3.2. Thrombin Increases Oxidative Stress within Atrial Endothelial Cells

Since reactive oxygen species (ROS) are strong inducers of senescence [15], experiments were performed to determine whether oxidative stress could be involved in thrombin-induced premature senescence using DHE. Indeed, thrombin (1 U/mL) increased the level of ethidium fluorescence in ECs (Figure 2A). The source of ROS was further characterized using inhibitors of major vascular sources of ROS including NADPH oxidase (VAS-2870), cyclooxygenases (COXs, indomethacin, INDO), COX-1 (SC-560), COX-2 (NS-398), the mitochondrial respiration complex (MIT INH), and the antioxidant N-acetylcysteine (NAC). All pharmacological tools blunted the thrombin-induced formation of ROS (Figure 2A). Similarly, NAC, VAS-2870, INDO and MIT also prevented the thrombin-induced SA-β-gal activity (Figure 2B). These findings suggest that NADPH oxidase, COXs and the mitochondrial respiration complex contribute to thrombin-induced oxidative stress and senescence in atrial ECs. In addition, Western blot analysis indicated that thrombin up-regulated COX-2, but not COX-1, in atrial ECs (Figure 3). Altogether these data suggest that thrombin induces a pro-inflammatory response in atrial ECs.

### 3.3. Thrombin Induces Endothelial Dysfunction Associated with Pro-Thrombotic and Pro-Inflammatory Patterns

Because endothelial senescence is characterized by endothelial dysfunction and a reduced formation of nitric oxide NO [15], the effect of thrombin on endothelial nitric oxide synthase (eNOS) protein expression was examined. A decrease in eNOS expression level was evidenced when ECs were exposed to thrombin (3 U/mL) to the same extent than that induced by AngII (100 nM, Figure 4A). The expression of vascular cell adhesion molecule (VCAM)-1 and TF was also examined to evaluate pro-inflammatory and pro-thrombotic patterns associated with ECs senescence. Concentration-dependent increases in VCAM-1 and TF expression levels were observed in atrial ECs in response to thrombin and to AngII (Figure 4B,C).

### 3.4. Thrombin Increases Pro-Fibrotic and Pro-Remodeling Responses

Since atrial remodeling is a key feature of AF [7], we investigated the link between thrombin-induced ECs senescence, TGF-β and the metalloproteinases MMP-2 and MMP-9. Previous studies have emphasized that TGF-β known to be involved in extracellular matrix remodeling is up-regulated in AF [21]. Thrombin induced a concentration-dependent increase in the expression level of TGF-β (Figure 5A). In addition, experiments were performed to determine the impact of thrombin on the expression level of active MMP-2 and 9 [22]. As depicted in Figure 5B,C, thrombin increased the active MMP-2 and 9 expression levels to a greater extent than AngII in atrial ECs.

### 3.5. Critical Role of the Local Angiotensin System in Thrombin-Induced Oxidative Stress in Atrial Endothelial Cells (ECs)

Since many reports have underscored that endothelial senescence is critically dependent on the redox activation by the local angiotensin system [15], experiments were performed to examine the interplay between thrombin and the local angiotensin system. We found that thrombin increased the expression level of ACE and AT1R in atrial ECs (Figure 6A,B). To determine whether AngII contributes to thrombin-induced oxidative stress, we evaluated the effect of losartan (AT1R antagonist) and perindoprilat (ACE inhibitor). Both AT1R blockade and ACE inhibition significantly blunted the thrombin-dependent induction of oxidative stress and cellular senescence (Figure 6C,D).

## 4. Discussion

The salient findings of the present study are as follows: (i) thrombin induces oxidative stress and premature atrial endothelial senescence, (ii) phenotypical changes associated with thrombin-induced atrial endothelial senescence comprise the acquisition of pro-thrombotic, pro-adhesive, pro-fibrotic and pro-remodeling patterns, (iii) pro-senescence response to thrombin is associated with an overexpression of both ACE and AT1R and is inhibited by perindoprilat and losartan, pointing to the involvement of the local angiotensin system. Altogether, these findings substantiate the view that thrombin, beyond its role in coagulation, is responsible for deleterious changes in atrial endothelial cells phenotype, thereby contributing to further worsen atrial function and paving the way to AF perpetuation.

Recent data have challenged the classical causality of AF by demonstrating that hypercoagulability per se induces pro-fibrotic and pro-inflammatory responses in adult atrial fibroblasts contributing to structural remodeling in the atria [6,7,9,23]. At the interface between blood and atrial tissue, the role of endothelium in response to thrombin activation remains conspicuously unexplored. Up to now, the understanding of endothelial dysfunction during AF was mainly based on studies performed on human umbilical vein ECs (HUVECs). To circumvent the possibility that endothelial function may vary according to the vascular bed, we developed an original model of primary cultures of porcine atrial ECs in order to test the hypothesis that thrombin could act as an effector of premature atrial senescence per se. The present findings indicate that thrombin, at concentrations achieved during vascular injury associated to thrombus formation [24], induced oxidative stress-dependent premature ageing of atrial ECs and enhanced thrombogenicity. Using a pharmacological approach, we evidenced that thrombin-induced oxidative stress involved several sources of ROS including NADPH oxidase, cyclooxygenases, and the mitochondrial respiratory chain. These pro-oxidant sources have also been involved in replicative senescence in coronary artery ECs [15]. Moreover, both NADPH oxidase and mitochondria were shown to be involved in endothelial MPs-mediated oxidative stress in mouse aortic ECs [25]. Interestingly, besides thrombin, activated Factor Xa, another mediator of hypercoagulability, has recently been reported in other cell lineages (HUVECs) to induce endothelial senescence [26].

Among the putative direct effect of thrombin on ECs, its impact on eNOS expression remains controversial. Indeed, previous reports have suggested that thrombin activates PAR-1 and stimulates eNOS with subsequent NO formation and vascular relaxation [27,28]. By contrast, in the setting of paroxysmal AF, Akar et al. have demonstrated that the acute onset of AF in humans is associated within minutes with thrombin generation and concomitant platelet aggregation together with a swift decrease in NO formation as a marker of endothelial dysfunction [29]. Likewise, in rabbits, short-term paroxysmal AF was evidenced to induce oxidative stress, thrombin generation and concomitant endothelial dysfunction [30]. Consistent with these findings, we could establish that thrombin significantly decreased eNOS expression in atrial ECs.

Another characteristic of senescent atrial ECs is the acquisition of a pro-thrombotic pattern, as evidenced by the enhanced expression of TF. Therefore, it is likely that at the senescent atrial endothelial surface blunted NO-mediated inhibition of platelet aggregation and enhanced TF expression, are both contributing to the acquisition of a pro-thrombotic pattern, and ultimately leading to thrombin generation feeding an amplification loop that further favors endothelial senescence.

The relevance of these findings was, for instance, emphasized by the demonstration that hypercoagulability promotes the development of AF in transgenic mice and in goats with persistent AF [9]. Nevertheless, independently of coagulation, recent data have highlighted that PAR-1 activation could also contribute to perpetuate AF by inducing inflammatory burden, atrial remodeling and, in term, alteration of electrophysiological characteristics of pulmonary vein and/or left atrium tissue [7,8,9]. The major importance of this pathway is corroborated by the fact that pro-remodeling effects of thrombin could be blunted by dabigatran, a direct thrombin inhibitor. In isolated rat atrial fibroblasts, a recent report by Spronk et al. depicted that thrombin enhanced the phosphorylation of the pro-fibrotic signaling protein kinase B and Extracellular signal-regulated kinase pathways and increased the expression of TGF-β and the pro-inflammatory factor monocyte chemo-attractant protein-1 [31]. The present findings extend these data by demonstrating that atrial ECs, when subjected to pro-senescent stimuli, constitute an important source of bio-effectors involved in inflammatory cell infiltration, fibrosis and extracellular matrix proteolysis.

The close relationship between AF, thrombogenesis and inflammation has been evidenced especially by increased monocyte-platelets-aggregates in patients presenting with AF [32]. Because endothelial dysfunction is associated with the expression of various cytoadhesins or selectins together with the production of proinflammatory chemokines, we investigated the impact of thrombin-induced endothelial senescence on VCAM-1 expression. VCAM-1 is a sialoglycoprotein expressed by cytokine-activated vascular endothelium that mediates cell adhesion and transendothelial diapedesis, which was recently evidenced as a potent marker of post-operative AF [33]. We evidenced here that thrombin is a potent inducer of VCAM-1 expression by senescent atrial ECs which may promote inflammatory cells infiltration within atrial tissues. Such an observation corroborates previous findings, demonstrating that thrombin induces endothelial NF-κB-dependent expression of ICAM-1 and VCAM-1 in SVEC4 mouse ECs [10]. Other data have underlined that both inflammation and oxidative stress are responsible for electrical and structural changes that promoted increased automaticity and autonomic dysfunction leading to an increased risk of AF [33,34].

In humans, senescence markers such as SA-β-gal activity and p16 were positively correlated with the extent of atrial fibrosis [12]. An important determinant of structural changes occurring during AF is represented by the pro-fibrotic cytokine, TGF-β. Recent data obtained with transgenic goat have shown that cardiac overexpression of TGF-β1 leads to increased fibrosis within the atria tissue and favors not only P wave prolongation but also AF vulnerability [35]. Other reports have previously established that mice overexpressing TGF-β1 developed a pronounced atrial fibrosis and an increased susceptibility to AF following rapid atrial pacing [36]. However, the impact of thrombin on endothelial TGF-β expression remains controversial. The significant up-regulation of TGF-β by thrombin in atrial ECs reported here challenges previous reports by Tang and coworkers describing a down-regulation of TGF-β signaling on aortic or venous human ECs stimulated by thrombin [37]. By contrast, recent data by Altieri et al. underlined that sustained stimulation by thrombin induced the synthesis of TGF-β by human atrial fibroblasts, which was counteracted by dabigatran [6]. Collectively, these novel data reinforce the view that atrial tissues, including atrial ECs, constitute a potent source of pro-fibrotic TGF-β when exposed to thrombin.

The MMPs represent other important regulators of structural remodeling associated to AF. For instance, it was recently established that fibrotic remodeling, as assessed by total collagen in the left atrium, is positively correlated to pro-fibrotic cytokines but also to MMP-2 and 9 in patients presenting AF [38,39]. The striking increase of atrium content in MMP-2 and 9 described in AF patients contrasting with the normal content of MMP-1 and 3 emphasizes the paradigm that MMP-2 and 9 are key effectors of matrix remodeling during AF [40]. Additionally, the thrombin-dependent up-regulation of active MMP-2 and 9 reported here is in line with former studies performed on rat cardiac fibroblasts [41]. By contrast, in human atrial fibroblasts, thrombin was found to mediate opposite effects by decreasing MMP-2 activity [6].

Although species dissimilarities in the nature of atrial tissue response to thrombin appear likely and hamper general extrapolation, our findings suggest that the senescent atrial ECs may participate to extracellular remodeling when subjected to thrombin stimulation. Previous data have highlighted that enhanced fibrosis, as a consequence of TGF-β signaling, paralleled the increase in atrial collagen deposition and alterations in extracellular matrix remodeling by MMPs. These structural alterations may pave the way to perpetuation of arrhythmia because fibrosis constitutes a primary mechanism in the disruption of connectivity between myocytes and impairs normal electrical conduction, thereby decreasing the wavelength of reentry [32].

Finally, another important determinant of structural changes associated to AF is represented by the angiotensin system [32]. Epidemiological studies have extensively demonstrated the association between AF and hypertension or elevated AngII [42]. Because AngII is a potent inducer of senescence in atrial endothelium and since ECs express high levels of ACE promoting AngII formation, we tested the hypothesis that the local angiotensin system may be involved in the pro-senescent effect of thrombin. We could establish that thrombin induced ACE and AT1R expression in atrial ECs. Furthermore, involvement of the local angiotensin system was emphasized by the demonstration that losartan, an AT1R antagonist, and also perindoprilat, an ACE inhibitor, blunted thrombin-induced oxidative stress and senescence. Altogether, these findings highlight a pivotal role of the local angiotensin system in the thrombin-mediated induction of premature endothelial senescence via activation of AT1R. Collectively, our data substantiate a new paradigm linking thrombin and AF to each other, in a vicious amplification loop where AF favors thrombin generation and thrombin per se, by inducing premature senescence of the atrial ECs, and shifts their phenotype towards pro-thrombotic, pro-inflammatory, pro-fibrotic and pro-remodeling patterns, promoting deleterious atrial remodeling and AF maintenance.

In conclusion, the present findings indicate that thrombin promotes premature ageing and senescence of atrial ECs and may pave the way to structural changes of the underlying atrial tissue by an up-regulation of the local angiotensin system and by promoting pro-inflammatory, pro-fibrotic and pro-remodeling responses. They further suggest that targeting the angiotensin system may be of interest to delay thrombin-induced endothelial atrial senescence.

## Figures and Tables

**Figure 1 jcm-08-01570-f001:**
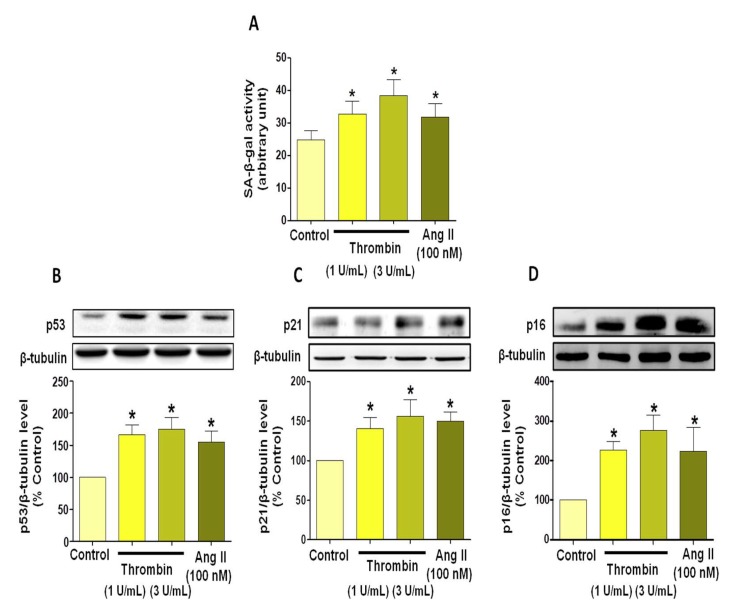
Thrombin and angiotensin II (AngII) induce senescence in atrial endothelial cells (ECs) at passage 1 and are associated with an up-regulation of major cell cycle regulatory proteins: p53, p21 and p16. Atrial ECs were either untreated or exposed to thrombin (1 or 3 U/mL) or AngII (100 nM) for 24 h before determination of senescence by SA-β-galactosidase (SA-β gal) activity (**A**) and protein expression level of (**B**) p53, (**C**) p21 and (**D**) p16 by Western blot analysis. Results are presented as representative immunoblots (upper panels), and corresponding cumulative data (lower panels) and are shown as mean ± SEM of *n* = 3–4 different experiments. **P* < 0.05 versus respective control.

**Figure 2 jcm-08-01570-f002:**
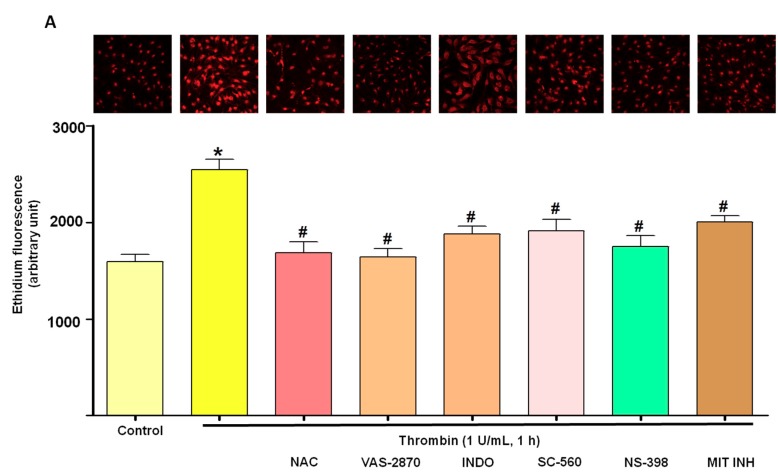
Thrombin induces oxidative stress promoting senescence in atrial ECs. (**A**) Atrial ECs were either untreated or exposed to N-acetylcysteine (NAC, an antioxidant), VAS-2870 (VAS, NADPH oxidase inhibitor), indomethacin (INDO, COX inhibitor), SC-560 (COX-1 inhibitor), NS-398 (COX-2 inhibitor) or a mitochondrial inhibitory complex (MIH; rotenone; KCN; myxothiazole) before the addition of thrombin (1 U/mL, 1 h) and dihydroethidium (DHE) to determine the level of oxidative stress by confocal microscopy. Upper panels represent ethidium staining and lower panel corresponds to cumulative data. (**B**) Atrial ECs were either untreated or exposed to NAC, VAS, INDO or MIH before the addition of thrombin (1 U/mL, 24 h) and subsequent determination of SA-β-gal activity using flow cytometry. Results are shown as mean ± SEM of *n* = 3–4 different experiments. **P* < 0.05 versus respective control, ^#^*P* < 0.05 versus thrombin-treated atrial ECs.

**Figure 3 jcm-08-01570-f003:**
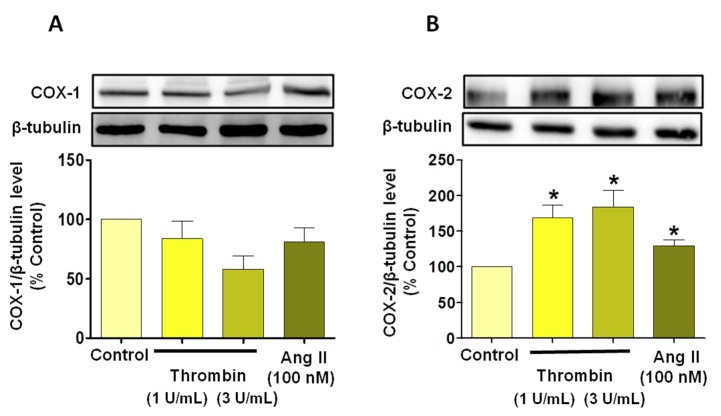
Thrombin and AngII induce the expression of cyclooxygenase-2 in atrial ECs. Atrial ECs were either untreated or exposed to thrombin (1 or 3 U/mL) or AngII (100 nM) before determination of the expression level of (**A**) COX-1 and (**B**) COX-2, as assessed by Western blot analysis. Results are shown as representative immunoblots (upper panels) and corresponding cumulative data (lower panels) and shown as mean ± SEM, *n* = 3–4, **P* < 0.05 versus respective control.

**Figure 4 jcm-08-01570-f004:**
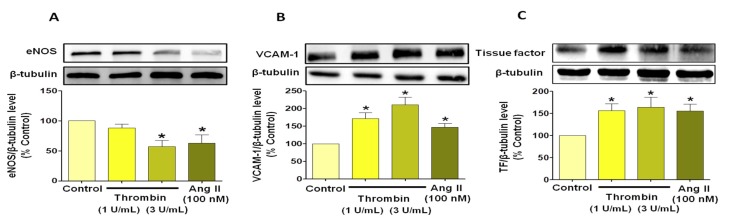
Thrombin- and AngII-induced senescence promotes pro-inflammatory and pro-coagulant phenotype in atrial ECs. Atrial ECs were untreated or exposed to either thrombin (1 or 3 U/mL) or AngII (100 nM) before determination of the expression level of target proteins, as assessed by Western blot analysis. Results are shown as representative immunoblots (upper panels) and corresponding cumulative data (lower panels) and shown as mean ± SEM, *n* = 3–4, **P* < 0.05 versus respective control.

**Figure 5 jcm-08-01570-f005:**
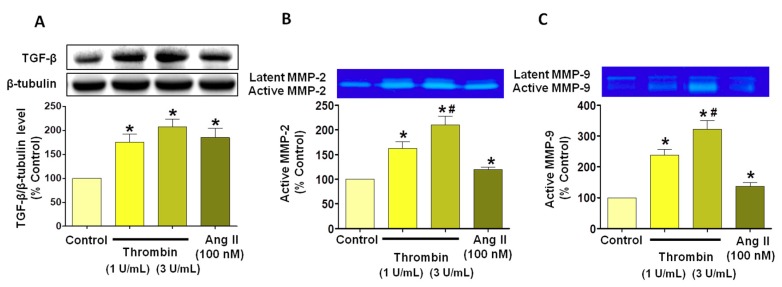
Thrombin- and AngII-induced senescence promotes pro-fibrotic phenotype in atrial ECs. Atrial ECs were untreated or exposed to either thrombin (1 or 3 U/mL) or AngII (100 nM) before determination of the expression level of TGF-β (**A**) as assessed by Western blot analysis, and MMP-2 and 9 activities by zymography (**B, C**). Results are shown as representative immunoblots or gelatinolytic activity (upper panels) and corresponding cumulative data (lower panels) and shown as mean ± SEM, *n* = 3–4, **P* < 0.05 versus respective control.

**Figure 6 jcm-08-01570-f006:**
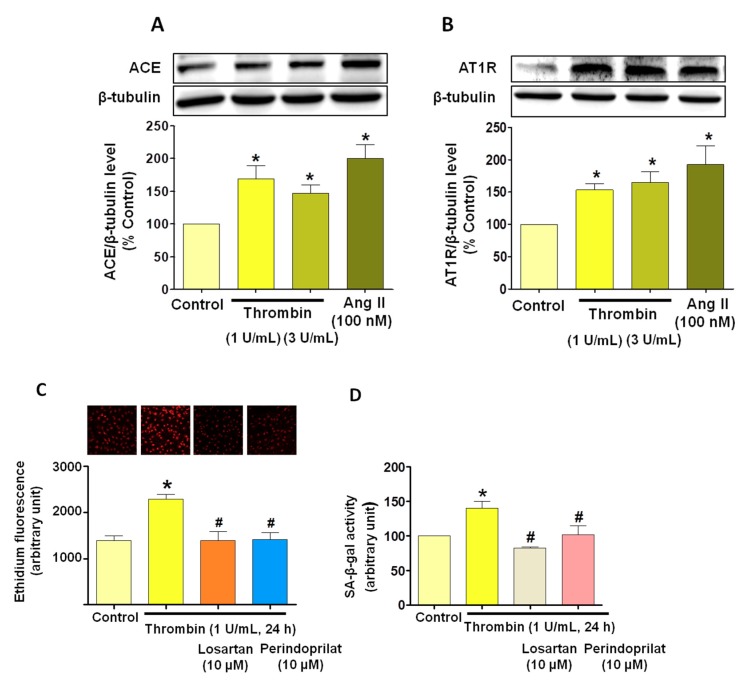
Thrombin-induced senescence promotes up-regulation of the local angiotensin system in atrial ECs. Atrial ECs were either untreated or exposed to losartan (AT1R antagonist) or perindoprilat (ACE inhibitor) before the addition of thrombin (1 U/mL) or AngII (100 nM) for 24 h, and the subsequent determination of the expression level of target proteins (**A, B**) as assessed by Western blot analysis, (**C**) oxidative stress by confocal microscopy, and (**D**) SA-β-gal activity using flow cytometry. Results are shown as representative immunoblots (upper panels) and corresponding cumulative data (lower panels) and shown as mean ± SEM, *n* = 3–4, **P* < 0.05 versus respective control, ^#^*P* < 0.05 versus thrombin-treated atrial ECs.

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
