# Peer review of "Thrombin Induces Angiotensin II-Mediated Senescence in Atrial Endothelial Cells: Impact on Pro-Remodeling Patterns"

_jcm, 2019, doi:10.3390/jcm8101570_

Round 1
Reviewer 1 Report
The manuscript entitled “Thrombin induces angiotensin II-mediated senescence in atrial endothelial cells: Impact on pro-remodeling patterns” assesses the potential of thrombin to promote premature senescence of endothelial cells (ECs) isolated from porcine atrial tissue. The topic is of clinical interest and appeals to the readership in cardiovascular medicine.
The introduction addresses prior evidence about the role of thrombin in atrial fibrillation through activation of PAR. It is encouraged to also include background knowledge of senescence associated secretory phenotype.
Page 4, Methods: “Premature atrial ECs senescence was induced at passage 1 by incubating cells with either thrombin (1 or 3 U/ml) or AngII (100 nM) for 24h in serum-free medium 15 h after seeding.” Please provide more information to support that these settings reflect clinically relevant concentrations of thrombin and AngII exposed to ECs.
Page 9, Figure 2: For the sake of completeness, it is suggested to evaluate the concentration-dependent increase in SA-β-gal activity of atrial ECs in response to thrombin by adding 3 U/mL to the results. Was the response to thrombin 3 U/mL prevented by NAC, VAS-2870, INDO, and MIT INH?
Page 15, Discussion: “Interestingly, besides thrombin, activated Factor Xa, another mediator of hypercoagulability, has recently been reported in other cell lineages (HUVECs), to induce endothelial senescence.” The study by Sanada et al. reveals that FXa induced EC senescence via stimulation of both PAR1 and PAR2, which activates the IGFBP-5-EGR-1-p53 pathway. How do the present results compare with these findings? Was IGFBP-5 involved in EC senescence induced by thrombin?
Author Response
The manuscript entitled “Thrombin induces angiotensin II-mediated senescence in atrial endothelial cells: Impact on pro-remodeling patterns” assesses the potential of thrombin to promote premature senescence of endothelial cells (ECs) isolated from porcine atrial tissue. The topic is of clinical interest and appeals to the readership in cardiovascular medicine.
First of all, we would like to thank the reviewer for its careful reading of our manuscript and his relevant comments. We think that the modifications in the light of these comments have improved the global quality of our manuscript.
The introduction addresses prior evidence about the role of thrombin in atrial fibrillation through activation of PAR. It is encouraged to also include background knowledge of senescence associated secretory phenotype.
Answer: We thank the reviewer for this suggestion. Some background with new references on the current knowledge concerning the secretory phenotype associated to senescence has now been added in the introduction.
Page 4, Methods: “Premature atrial ECs senescence was induced at passage 1 by incubating cells with either thrombin (1 or 3 U/ml) or AngII (100 nM) for 24h in serum-free medium 15 h after seeding.” Please provide more information to support that these settings reflect clinically relevant concentrations of thrombin and AngII exposed to ECs.
Answer: We thank the reviewer for these comments. Since it has been reported to induce senescence in ECs [1, 2], AngII was used as a “positive control” in our experiments. The concentration of 100 nanomolar is classically used in vitro to induce submaximal AT1R stimulation. Moreover, in our hypothesis, local renin angiotensin system serves as an amplifier of the effect of thrombin (as shown by the effect of specific inhibitors). Local concentrations of AngII are virtually impossible to measure but have however been estimated to reach the nanomolar range, which corresponds to the affinity of AngII receptors. It is assumed that angiotensin in blood, which approximates few dozen picomolar, may only represent overflow products of a locally acting system.
Concerning thrombin, its local generation in pro thrombotic conditions, as those encountered in atrial fibrillation for example, can reach concentration of hundred nanomolar which are equivalent to several units per mL [3]. We now added this information and the appropriate reference in the text (Materials and Methods, Isolation and culture of atrial endothelial cells).
Page 9, Figure 2: For the sake of completeness, it is suggested to evaluate the concentration-dependent increase in SA-β-gal activity of atrial ECs in response to thrombin by adding 3 U/mL to the results. Was the response to thrombin 3 U/mL prevented by NAC, VAS-2870, INDO, and MIT INH?
Answer: Noteworthy, there were no statistical differences between the effect of 1 and 3U/mL thrombin in term of senescence induction (SA-β-gal activity). For sake of homogeneity with other experiments we performed SA-β-gal activity test using 1U/ml.
Page 15, Discussion: “Interestingly, besides thrombin, activated Factor Xa, another mediator of hypercoagulability, has recently been reported in other cell lineages (HUVECs), to induce endothelial senescence.” The study by Sanada et al. reveals that FXa induced EC senescence via stimulation of both PAR1 and PAR2, which activates the IGFBP-5-EGR-1-p53 pathway. How do the present results compare with these findings? Was IGFBP-5 involved in EC senescence induced by thrombin?
Answer: We thank the reviewer for this comment. We did not investigate specifically the involvement of the pathway IGFBP-5 / EGR-1 / P53 in thrombin-induced senescence in our model. We believe that the investigations of specific signaling were beyond the scope of our present work. This suggestion opens exciting perspectives for future investigations on the molecular mechanism implicated in cellular senescence induced by FXa vs thrombin in atrial cells. These questions would deserve a specific dedicated work.
References
Shan, H., X. Bai, and X. Chen, Angiotensin II induces endothelial cell senescence via the activation of mitogen-activated protein kinases.Cell Biochem Funct, 2008. 26(4): p. 459-66. Abbas, M., et al., Endothelial Microparticles From Acute Coronary Syndrome Patients Induce Premature Coronary Artery Endothelial Cell Aging and Thrombogenicity: Role of the Ang II/AT1 Receptor/NADPH Oxidase-Mediated Activation of MAPKs and PI3-Kinase Pathways.Circulation, 2017. 135(3): p. 280-296. Mann, K.G., K. Brummel, and S. Butenas, What is all that thrombin for?J Thromb Haemost, 2003. 1(7): p. 1504-14.

Reviewer 2 Report
The authors examined the potential of thrombin to promote premature atrial endothelial cells (ECs) senescence in 24 Primary ECs from porcine atrial tissue. In the presented work the authors concluded that Thrombin promotes premature ageing and senescence of atrial ECs and may pave the way to deleterious remodeling of atrial tissue by a local up-regulation of the angiotensin system and by promoting pro-inflammatory, pro-thrombotic, pro-fibrotic and pro-remodeling responses.
Author Response
We would like to thank the reviewer for his careful reading of our manuscript and for his positive evaluation of our work.